# Microglia and Alzheimer’s Disease

**DOI:** 10.3390/ijms232112990

**Published:** 2022-10-27

**Authors:** Stefania Merighi, Manuela Nigro, Alessia Travagli, Stefania Gessi

**Affiliations:** Department of Translational Medicine and for Romagna, University of Ferrara, 44121 Ferrara, Italy

**Keywords:** Alzheimer’s disease, microglia, neuroinflammation

## Abstract

There is a huge need for novel therapeutic and preventative approaches to Alzheimer’s disease (AD) and neuroinflammation seems to be one of the most fascinating solutions. The primary cell type that performs immunosurveillance and helps clear out unwanted chemicals from the brain is the microglia. Microglia work to reestablish efficiency and stop further degeneration in the early stages of AD but mainly fail in the illness’s later phases. This may be caused by a number of reasons, e.g., a protracted exposure to cytokines that induce inflammation and an inappropriate accumulation of amyloid beta (Aβ) peptide. Extracellular amyloid and/or intraneuronal phosphorylated tau in AD can both activate microglia. The activation of TLRs and scavenger receptors, inducing the activation of numerous inflammatory pathways, including the NF-kB, JAK-STAT, and NLRP3 inflammasome, facilitates microglial phagocytosis and activation in response to these mediators. Aβ/tau are taken up by microglia, and their removal from the extracellular space can also have protective effects, but if the illness worsens, an environment that is constantly inflamed and overexposed to an oxidative environment might encourage continuous microglial activation, which can lead to neuroinflammation, oxidative stress, iron overload, and neurotoxicity. The complexity and diversity of the roles that microglia play in health and disease necessitate the urgent development of new biomarkers that identify the activity of different microglia. It is imperative to comprehend the intricate mechanisms that result in microglial impairment to develop new immunomodulating therapies that primarily attempt to recover the physiological role of microglia, allowing them to carry out their core function of brain protection.

## 1. Introduction 

Amyloid plaques and neurofibrillary tangles (NFTs) are two characteristics of Alzheimer’s disease (AD), a neurodegenerative condition affecting more than 50 million individuals in the world expected to rise to 139 million by 2050 [1]. Even though the disease’s enormous frequency, the pathogenic processes are still not completely understood. While preventive and therapeutic solutions are still to be offered, treatments that just address symptoms have been established [2]. Gaining more insight into AD pathophysiology and identifying possible therapeutic targets that could be used as preventive or curative measures against the illness are essential if we are to decrease prevalence and morbidity globally. The primary mechanisms assumed to be implicated in AD pathology up until this point have been the amyloid and tau hypothesis. The amyloid hypothesis continues to be the most widely accepted explanation for AD pathology because of the substantial evidence for the crucial role that amyloid-beta (Aβ) plays in the disease’s pathology over the past few decades. However, the hypothesis falls short of providing a comprehensive explanation for the disease’s underlying causes. The failure of Aβ-targeted immunotherapies demonstrates more flaws in the amyloid theory. No cognitive enhancement was seen [3], despite a drop in Aβ levels in response to these treatments.

The recent FDA approval of aducanumab (Aduhelm^®^, Biogen, Cambridge, MA, USA), which targets amyloid, was strongly debated as having fallen short in the EMERGE and ENGAGE studies in demonstrating its ability to improve cognitive function in AD patients. Biogen abruptly terminated phase III clinical trials of the monoclonal antibody that targets Aβ, citing considerable favorable trends in the data and the relevance of applying the treatment to AD. Phase III clinical trial data were further analyzed, and it was discovered that there was no therapeutic effect on cognitive performance [4]. In addition to the amyloid cascade concept, research in animals or clinical trials have shown that tau-dependent neurodegeneration is started by Aβ while Tau raises amyloid accumulation [5,6]. Although the amyloid theory is thought to be the cause of AD, especially early-onset AD, there has recently been speculation that tau pathology may be the cause of late-onset AD, and this needs more research [7]. The synergy of Aβ and Tau with participants of other pathogenic elements, such as neuroinflammation, results in neuronal death, synaptic dysfunction, and cognitive loss that are implicated in the progression of symptoms of AD [8]. Recent years have seen an increase in interest in the development of disease-modifying therapies that specifically target the tau pathology due to the lack of association between the degree of cognitive dysfunction and the efficacy of Aβ-targeted medicines in clinical trials. Tau aggregation, tau post-translational changes, and cytoskeletal stabilization are among the potential treatments for tauopathy [9].

In pathological conditions Tau is more prone to aggregation and has a lower affinity for microtubules, which has an impact on neuronal plasticity. Indeed, approaches for tackling tau include regulating Tau phosphatases and kinases, inhibiting Tau accumulation, via Tau vaccinations, and stabilizing microtubules [10]. Unfortunately, in clinical trials, the majority of these attempts stayed a failure.

For instance, phase III studies of the tau accumulation blocker TRx0237 [11] revealed its unsuccess. Phase I and II clinical trials are currently being conducted to examine active tau-targeted vaccines (ACI35 and AADvac-1) as well as inactive tau-targeted vaccines (RG6100 and ABBv-8E12) [12,13]. The only passive immunization investigated in phase III clinical trials for those with mild to severe AD is intravenous immunoglobulin, although it does not meet the primary endpoints [13]. Other therapeutic approaches for AD having tau as target, including microtubule stabilization and kinase and phosphatase regulation, have recently received attention in preclinical studies. Misgivings about the efficacy of Aβ-targeting and tau pathological therapeutics may highlight the potential advantage of looking for a novel strategy that integrates these two pathologies in order to prevent and postpone disease development.

One frequent trait of AD patients is neurodegeneration. It first manifests in the temporal lobe, decreasing patients’ capacity to retain short-term memory, and then develops in the parietal lobe, impairing long-term memory formation as well [8].

As a result, neurodegeneration is associated with cognitive impairment in AD patients, emphasizing the significance of this association for comprehending the symptomatic progression of cognitive decline. Targeting neuronal degradation would be the ideal focus of AD treatment to stop the disease’s development. The lack of clinically authorized medications to prevent or stop neurodegeneration is mostly a result of our incomplete understanding of the underlying mechanisms. According to a study on autosomal dominant AD, pathophysiological changes, such as amyloid plaques, the quantity of tau protein in the CSF, and brain shrinkage, start at least two decades before the development of clinical symptoms [14]. Numerous studies on the amyloid and tau alterations in AD have sped up the development of preventive and therapeutic approaches focused on Aβ and tau; however, this focus may overshadow the significance and opportunities of another promising target, neuroinflammation, which has recently gained attention [15,16,17].

Neuroinflammation is described by immune system stimulation in the CNS, which results in greater release of chemical mediators. Initially, neuroinflammation was thought to be a secondary reaction, followed by the amyloid cascade and neurofibrillary tangles. However, recent work suggests that neuroinflammation is also a causal agent in AD development, and immune system alteration occurs earlier before the initiation of AD symptoms. It is proposed that inflammation may influence generation of this pathology separately by amyloid and tau pathways and cooperate with them, further perpetuating the vicious cycle [18].

There are extremely few choices for treating AD. Only four medications are currently approved for the treatment of AD by the European Medicines Agency (EMA) and the Food and Drug Administration (FDA). These include memantine, a selective and noncompetitive NMDA receptor antagonist, and three medications from the class of acetylcholinesterase (AChE) inhibitors (donepezil, rivastigmine, and galantamine) [17,19,20,21]. AChE inhibitors raise ACh contents in the synaptic space and in part enhance cognitive function and quality of life in patients [2]. Memantine improves NMDA receptor activity and exerts neuroprotection by lowering calcium ion intracellular influx [20,22]. There is a huge need for novel therapeutic and preventative approaches to AD and one of the most intriguing strategies to solving this problem appears to be neuroinflammation.

Macrophages and microglia typically move to the location of the injury as the initial reaction to CNS injury [23,24]. Microgliosis is the term for changes in the morphology and increased expression of microglia [25]. In order to aid in remyelination, oligodendrocyte cells are also brought to the injured area [26]. The final element is the increased expression of astrocytes, which are the primary cells of the glial response and undergo concurrent structural alterations [26,27]. Astrogliosis is the name for these alterations in astrocytes. Therefore, in the AD pathophysiology, neuroinflammation modulates the cognitive impairment and memory loss. A reactive gliosis may be a crucial factor in the formation and progression of AD. Recent genomic and transcriptome studies have supported the idea that pathways associated to microglia are essential for the risk and pathogenesis of AD [10]. It’s interesting to note that recent research has shown that microglia accumulate close to amyloid-beta plaques in the AD brain and increase disease severity [28,29,30]. Therefore, the topic of this review is to offer an overview on microglia involvement in neuroinflammation and AD pathology.

## 2. Microglia Physiology

Microglia are mononuclear phagocytes thought to be part of the immune system that live permanently in the CNS [24,31,32]. Their function is comparable to that of blood macrophages, which have phagocytic and inflammatory properties. Pio del Rio Hortega described microglial cells between 1919 and 1927 [33,34,35,36]. 

A variety of stimuli can activate microglia, including bacterial cell wall lipopolysaccharides (LPS), pesticides (e.g., paraquat), misfolded proteins (e.g., Aβ, ɑ-synuclein), and air pollutants [37,38]. Toll-like receptors (TLRs), NOD-like receptors (NLRs), receptors for advanced glycation end products (RAGE), scavenger receptors, formyl peptide receptors, complement receptors, and Fc receptors are all involved in the microglial response [39]. All of them are among the immune pattern recognition receptors (PRRs) expressed by microglia [39,40]. PRRs identify either extracellular pathogenic substances called pathogen-associated molecular patterns (PAMPs) or endogenous host-derived molecules called damage-associated molecular patterns (DAMPs) [41,42]. PAMPs and DAMPs have different responses to infection. PAMPs cause an antimicrobial response and inflammation, whereas DAMPs activate surface receptors that recognize signals released from damaged cells in response to CNS injuries such as trauma, hypoxia, and neurodegenerative diseases [43]. A number of intracellular cascades, kinases, and downstream transcription factors are triggered when PAMP/DAMPs engage microglial PRRs, resulting in the production of inflammatory mediators as well as other cellular responses [42,43].

For a number of years, it was believed that microglia activation was harmful; these cells were referred to as reactive microglia [44]. Because diverse phenotypes are produced when microglia are activated, it is now understood that these cells are crucial for neuroprotection [45]. In physiological conditions, the microglia are dormant (M0) and exhibit ramified processes [46]. Findings, however, transformed our understanding of the role of microglia by demonstrating that these cells are constantly scanning the brain for any changes to cerebral homeostasis and never rest [47]. More specifically, these studies demonstrated that microglia act as guardians to evaluate and scan the milieu in their immediate surroundings for any changes in brain homeostasis [47]. Any harmful circumstance causes cell activation, which, as occurs in macrophages, may result in distinct microglial phenotypes. The two primary activation phenotypes are M1 and M2, with M1 having a proinflammatory phenotype and M2 taking part in resolution of inflammation [48] (Figure 1). LPS and/or IFNγ both promote the M1-like microglia phenotype in vitro. Specifically, LPS activated TLR4, via TIR domain-containing adapter inducing IFN (TRIF) and myeloid differentiation primary response protein 88 (MyD88)-dependent pathways to trigger NF-kB, AP1, STAT5, and interferon regulatory factors (IRFs), among other pro-inflammatory transcription factors [49,50]. As for IFNγ, it binds to IFNγ receptors 1 and 2 (IFNγR1/2) and stimulates the JAK/STAT cascade, which enables STAT1 and other IRFs to be phosphorylated and translocated into the nucleus [51]. Pro-inflammatory cell membrane biomarkers including MHCII and the cluster of differentiation marker 86 (CD86) are upregulated as a result of transcription factors that are triggered by M1-like microglia [52,53]. They also increase the production of numerous pro-inflammatory mediators, such as tumor necrosis factor (TNF) and interleukins IL-1, IL-6, IL-12, IL-17, IL-18, and IL-23 cytokines, CCL12 and CXCL10 chemokines, and other pro-inflammatory mediators, reactive oxygen and nitrogen species (ROS and RNS), inducible nitric oxide synthase (iNOS), and cyclooxygenase-2 (COX-2) [53,54,55,56]. M1-like microglia are crucial for inducing the adaptive immune response and innate immunological responses to fight invading microorganisms [57]. Nevertheless, persistent activation under pathological circumstances is linked to neurotoxicity, oxidative stress, and neuroinflammation [57,58]. The release of anti-inflammatory and neurotrophic substances by M2 polarized microglia is frequently linked to tasks including immunological resolution and tissue repair [58,59,60]. These microglia can also take a “alternatively activated” or “acquired deactivation” state. The CNS has endogenous defensive systems that encourage tissue regeneration when brain homeostasis is disturbed as a result of brain damage or prolonged stress. Injured neurons emit a variety of anti-inflammatory cytokines, growth factors, and hormones such glucocorticoids that encourage the microglia in the area to change into a protective phenotype similar to M2 [61,62]. Anti-inflammatory cytokines such as IL-4, IL-10, IL-13, and TGF-β can activate M2 microglia. IL-4 and IL-13 enhance the alternative activation state and, in general, operate to inhibit M1 pro-inflammatory responses, such as TNFɑ, IL-6, and iNOS generation [63,64]. TGF-β, a multifunctional cytokine that promotes angiogenesis, immunoregulation, and tissue regeneration, induces the acquired deactivation state in combination with IL-10 [59]. M2 phenotypes are divided into three types: M2a, M2b, and M2c, which share biochemical functions but differ in activating stimulation, marker expression, and mode of action [55]. Therefore, microglia have the ability to dynamically switch between M1- and M2-like polarization states. Instead of using two separate states of activation, M1 and M2 indicate a spectrum of activation phenotypes that are continuous, and diverse phenotypic markers can coexist, suggesting many intermediate phenotypes [58]. In conclusion, the role of microglia in the healthy and pathological brain and the change between various phenotypic states are determined by the engaging mechanism, the damage’s time-frame, and the regulatory signaling molecules involved [65,66].

## 3. Microglia in AD

### 3.1. Biomarkers 

Microglia have been suggested as key participants in AD and cognitive decline may be induced at least in part by the increase of microglia, as demonstrated by overexpression of its specific biomarker IBA-1 [67]. IBA-1 and the fractalkine receptor CX3CR1, which is not only present in microglial cells and is also a receptor for the neuronal ligand CX3CL1, form an essential pathway for interplay between neurons and microglia [68]. Although every microglia expresses IBA-1, it would be desirable to employ other biomarkers to identify active cells, such as CD68 [69]. It would also be easier to grasp the findings if there was a biomarker that measured the quantity of activated microglia as well as overall microglia levels. Upregulation of IBA-1, biomarker for microglia [70], is also coupled to an increase of interleukin (IL)-1β, signal transducer and activator of transcription 3 (STAT3), nuclear factor kappa-light chain-enhancer of activated B cells (NF-κB), tumor necrosis factor alpha (TNF-α), and glycogen synthase kinase 3 beta (GSKβ). In addition, a decrease of other molecules, such as brain-derived neurotrophic factor (BDNF) and cAMP response element-binding protein (CREB) in activated cells is observed [71,72,73]. While IL-1β, STAT3, NF-kB, TNF-α, and GSK are linked to the activation of several pro-inflammatory pathways that would impair cognition and memory, CREB and cAMP play a critical role in cognitive performance and memory preservation [71,74,75] (Figure 2). Accordingly, the deletion of NLRP3, the master regulator of IL-1β, significantly improves cognitive impairment in AD mouse models, in addition to lowering Aβ-induced microglial activation in vitro, Aβ-deposition, or Tau pathology. Therefore, an adequate cellular homeostasis appears to depend on fine-tuning of NLRP3 inflammasome, that was shown to be abnormally activated in AD [76,77,78,79,80,81]. 

Microglia linked to Aβ plaque have been discovered in the brains of AD patients. Increased proliferation [82,83] and expression of inflammatory markers including CD36, CD14, CD11c, MHC-II, and iNOS [84,85], as well as indicators of the M1 phenotype (such as IL-1, MCP-1, MIP-1, IL-1, TNF, and IL-6), and chemokine receptors are all signs of activated microglia in AD (e.g., CCR3, CCR5). Nevertheless, a brand-new form of microglia phenotype known as “dark microglia” was discovered in circumstances such as prolonged stress, particularly in the APP/PS1 mice model of AD [86]. Notably, when the microglia were connected to amyloid deposits, they displayed a highly active phenotype with robust expression of CD11b and TREM2 and widespread surrounding of synaptic clefts [87]. This finding implies the presence of many different microglia phenotypes.

The “disease-associated microglia” (DAM) phenotype or the “microglial neurodegenerative” (MGnD) phenotype with APOE, TREM2, CSF1, CST7, and SPP1 marker genes was recently characterized as a specific disease-associated microglial phenotype in neurodegenerative illness [88]. Microglia with MGnD have the ability to phagocytose pathogens, toxic chemicals, dead or wounded neurons, as well as enhance inflammatory processes. Microglia can operate as a two-sided weapon that promotes AD due to their ongoing activation by a variety of triggers [88].

### 3.2. External and Intrinsic Signals Causing Microglia Dysfunction

It is interesting to report that different external and intrinsic signals may cause a dysfunctional reaction of microglia during AD. As a relevant example, the relationship between oxidative stress and brain function has been widely recognized for the past 20 years. In addition, transition metals play a crucial part in a number of biological events that result in free radicals, according to this line of research. Reactive oxygen species (ROS), which cause cellular compartments to be damaged and destroyed, are known to be produced in Fenton-type reactions that involve iron and copper [89]. A significant body of research has revealed the part that metals play in the development of the hazardous pathways that lie at the heart of AD. Metals, as possible AD drivers, have, however, been mostly ignored since the 1990s due to the amyloid hypothesis. In the brain, biometals including iron, copper, and zinc are tightly monitored. Their neurotoxic effects are not just a result of greater exposure, but also of homeostasis disruption and the compartmentalization of oxidative stress or excitotoxicity that follows. The alteration of brain metal metabolism is thought to result from both hereditary and non-genetic causes. Intake and release, storage, intracellular metabolism, and control are some of the levels at which it might also manifest [90]. We were able to confirm the interest in metallobiology of neurodegenerative diseases as originally presented by biochemical considerations [91] by discovering some many Loss-of-Function variants in genes encoding for regulatory proteins of metal metabolism. Understanding the role of metal homeostasis in brain functionality has been made possible by genetic investigations of hereditary metal metabolism diseases [92,93]. These studies have shed light on various altered pathways associated to metals that lead to brain dysfunction. In particular, they modify the redox state of the cellular environment, catalyze redox events physiologically, and disturb neural architecture [94], thus having demonstrated crucial roles in neurobiology. Iron, copper, and zinc, in particular, play important roles in a number of neurologic processes, such as energy metabolism, antioxidant defense, myelination, DNA synthesis, neuronal cytoskeleton integrity, and the production of neurotransmitters [95]. In addition, metal ions are the main industrial, and vehicular pollutants that put people’s health at risk by posing a serious threat to the environment. A production of Aβ is regulated, according to several studies [96,97,98], by metal ions including copper, iron, aluminum, and others. In senile plaques of AD patients, zinc and iron contents considerably rise and Aβ accumulation is greatly retarded by the treatment of metal chelator [99]. Through a number of mechanisms, including oxidative stress [100,101], redox active metal ions have an impact on Aβ-mediated neurotoxicity. In particular, several epidemiological studies have revealed that aluminum exposure is closely associated to neurodegenerative diseases including AD, PD, and amyotrophic lateral sclerosis (ALS) [102,103]. Acute aluminum exposure has been shown to harm the nervous system, whereas chronic aluminum exposure over time results in the aging and neurodegeneration of the nervous system [104]. Aluminum’s neurotoxic effects mostly show up as cognitive decline [105]. The learning and memory function in employees exposed to aluminum is significantly diminished, according to results from the World Health Organization’s (WHO) neurobehavioral core test [106,107]. Aluminum causes damage to nerve cells’ mitochondria [108], apoptosis and programmed necrosis [109,110], Aβ deposition [111], abnormal tau protein phosphorylation [112], changes in synaptic plasticity and glial activation. 

### 3.3. Signaling

Microglia activation happens before AD manifests, according to in vivo PET imaging studies [113]. Microglia perform preventive actions, including the removal of Aβ, the inhibition of tau hyperphosphorylation, and the production of neurotrophic factors, which delay the onset of AD symptoms [28,114,115,116,117]. However, when the illness worsens, the microglial responses change, and continued stimulation of the microglia causes neurodegeneration [118]. This supports a heterogeneous microglia activation reaction along the course of AD, because moderate stimulation results in a protective phenotype that is evident in the preclinical phases and excessive activation results in a neurotoxic phenotype that manifests during the clinical phase [113,118]. As a result, therapeutic strategies involving microglia now concentrate on addressing the first biochemical process prior to the beginning of pathology, such as treating individuals with moderate cognitive impairment who still haven’t shown signs of dementia. Amyloid load in the brain is regulated by the balance between Aβ production and clearance, and systems that control Aβ metabolism have been connected to the etiology of AD [119]. Indeed, microglia’s capacity to prevent plaque development by removing pathogenic Aβ is one of their well-known advantageous roles in the setting of AD [115]. As a protective strategy to support the first-line defense response and prevent further Aβ accumulation, Aβ controls a number of microglial actions, including chemoattraction, stimulation, and growth [120]. The association of Aβ with different microglial receptors, including TREM2, TLRs, CD36, class A1 scavenger receptors (SR-A1), and receptor for advanced glycation end products (RAGE), mediates Aβ phagocytosis/endocytosis in microglia [121]. Recent data have been obtained on the role played by TAM receptors tyrosine kinases in the detection and engulfment of Aβ plaques. Specifically, microglial identification, reaction, and phagocytosis of Aβ plaques are all dependent on the TAM system. Furthermore, dense-core plaque development is not inhibited by TAM-mediated microglial phagocytosis of Aβ material, instead it is promoted by it [122]. This may aid in understanding why most attempts to treat AD with drugs that disaggregate dense-core plaques but do nothing to affect the generation of Aβ peptides have ended in failure [123,124].

The exact kind of receptor that is activated and the structural shape of Aβ affect the microglia’s general reaction [120]. Cerebral pro-inflammatory cytokines in AD pathology raise microglial iNOS, which produces toxic NO [125,126]. Increased NO contents in the brain have been linked to neurotoxicity, alterations in mitochondrial function, and Aβ protein interaction, which promotes plaque aggregation [127]. Aβ activates a number of inflammatory pathways that result in the generation of pro-inflammatory cytokines, reactive oxygen and nitrogen species (ROS/RNS). Pro-inflammatory microglial responses are largely amplified by Aβ-mediated activation of the CD36/TLR4/TLR6 complex [128], and activation of RAGE also drives inflammatory responses in microglia and encourages oxidative stress in neurons [129]. Aβ stimulates the NLRP3 inflammasome, a cytoplasmic tricomplex made up of the effector protein caspase-1, an adapter protein called ASC, and a sensor protein called NLRP3, which together promote IL-1β release and cause neurodegeneration [16,130]. ASC released by active microglia further aids in plaque formation by causing Aβ to oligomerize and aggregate, and NLRP3 inflammasome stimulation can change the microglial phagocytosis function and increase Aβ accumulation [131,132]. Collectively, Aβ may trigger a number of neurotoxic and inflammatory signalling in microglia that can contribute to AD pathogenesis.

Aging-related microglial senescence can make microglia more sensitive to inflammatory stimuli and worsen Aβ disease [133]. Indeed, in AD pathology, while Aβ can cause persistent microglia activation, inflammation in the brain microenvironment can decrease Aβ clearance systems and encourage Aβ accumulation, thus promoting AD occurrence. For instance, pro-inflammatory cytokines, such as TNFɑ and IFNγ that are produced by microglia, might impair their capacity to digest Aβ and block the production of Aβ-degrading proteases, which can lead to plaque formation [134,135]. Data show that defective microglial clearance pathways contribute to Aβ buildup in the early stages of AD.

Indeed, it has been demonstrated that microglial Aβ-binding cognate receptors, including CD36, SR-A, and RAGE, are downregulated in old mice and AD brain, which points to an altered phagocytosis function with aging [136]. Furthermore, studies comparing isolated microglia cultured for 2 or 16 days revealed that older microglia have a lower response to Aβ in terms of phagocytosis and autophagy, as well as a decreased expression of TREM2 and an overexpression of the senescence-associated galactosidase [137]. In addition, it has been reported that fresh microglia might revitalize older microglia’s ability to remove amyloid plaques [138]. These results suggest that defective microglial Aβ clearance contributes to plaque accumulation with aging, suggesting that therapeutic intervention targeting age-associated microglial senescence may delay the onset and progression of AD. Conversely, a suitable approach for a novel cell transplantation therapy for AD may involve substituting older microglia with young functional phenotypes that present appropriate quantities of autophagic receptors. Recently, the replacement of microglia by circulation-derived myeloid cells in a mouse model of progressive dementia decreased cell loss and brain inflammation, enhanced motor function, and increased lifespan. The findings imply that this strategy may be beneficial in treating a number of neurological diseases [139].

Another function of microglia involves Tau regulation, being activated microglia near to NFTs in AD patients [140]. Microglia have the ability to ingest and destroy pathogenic tau particles, but they can also help tau pathology expand [141,142]. 

Tau accumulation precedes the presence of active microglia cells in experimental AD models [143,144], suggesting that microglial inflammation plays a role in aggravating tau aggregation and dissemination. This is supported by the use of well-known immunosuppressants, such as the anti-inflammatory agent minocycline, that regulate microglia activity and the mitigation of tau pathology in a mouse model of tauopathy and in AD patients [145,146]. Microglial inflammation has been found in studies to speed up NFT production [142,147]. The NLRP3 inflammasome complex can be engaged by tau absorption in microglia, which encourages tau seeding [130]. Tau activity can also be modified by activated microglia by post-translational changes. For instance, pro-inflammatory cytokines such as IL-1 and IL-6 that are generated by active microglia cause tau phosphorylation, which encourages the development of NFTs [148]. Tau is ubiquitinated by microglia, and ubiquitinated forms of tau are well integrated into exosomes, which can promote the releasing of tau seeds outside of cells [142]. Regulatory CX3CR1-CX3CL1 signaling can be disrupted by pro-inflammatory events in microglia [149], which could further cause prolonged microglial activation and support NFT development [48]. The breakdown of tau is mediated by proteasomal and autophagic systems, and there is indication that both clearance mechanisms are compromised in AD [150]. The receptors related to the interaction with tau are poorly understood, but it has been shown that tau interacts with the microglial CX3CR1 receptor, facilitating tau phagocytosis and internalization [151]. Nevertheless, tau and CX3CL1 are antagonistic to one another in the AD brain, and persistent CX3CR1/tau signaling can enhance pro-inflammatory and neurotoxic pro-inflammatory responses [152]. However, depending on the stage of the disease, microglia likely exert various impacts on tau pathology. The precise mechanism behind microglia’s dual roles is unclear. In conclusion, defects in tau clearance systems and neuroinflammatory mechanisms that encourage tau aggregation and propagation collectively contribute to microglia-induced tau disease. To fully comprehend the function and mechanism of the microglia-tau relationship at different stages of AD progression, more research is required.

## 4. Microglia, Astrocytes and Neurons Crosstalk in AD

To better understand the development of AD, it is important to consider how microglia interact with other glial and nerve cells. It is worth noting that, by regulating synaptic plasticity and neurotransmission, neurons and glia are known to work together to modulate cognitive activities [57].

Interestingly, recent research has indicated that glial cell dysregulation may play a significant role in the cognitive abnormalities and neurodegenerative processes found in AD. Microglial motility, astrocyte and microglia proliferation, and perhaps both of their phagocytic functions have been seen to be induced by astrocyte-microglia crosstalk [153,154]. Similar to microglia, astrocytes have also been discovered to be in a disease-associated condition, with elevated levels of glial fibrillary acidic protein (GFAP) and increased levels of genes related to inflammatory signaling and reactivity to toxic substances [155]. These toxic-reactive astrocytes, known as A1, have been demonstrated to be produced by activated microglia through pro-inflammatory cytokines, and they have been found to be more prevalent in the prefrontal cortex in AD patients [156]. Interestingly, reactive astrocytes and activated microglia are frequently identified close to the senile plaques of AD patients, indicating their critical role in the disease’s etiology [157]. Additionally, it has been noted that TNFα which has been demonstrated to be produced from microglia, macrophages, and neutrophils, is necessary for astrocytes to develop into A1 [156,158]. A rise in GFAP expression was seen in AD tissue after the loss of myelination of neurons [159]. There is an increase in astrocytes in the TgF344 rat model of AD, which may contribute to excitotoxicity and neuronal death by producing a co-agonist of the N-methyl-D-aspartate receptor (NMDAR) on neurons [160].

Importantly, after CNS insult or injury, astrocytes and microglia produce a number of signaling molecules (such as growth factors, neurotransmitters, cytokines, and chemokines), generating a bidirectional interaction for a tight reciprocal regulation [161,162]. Injured neurons specifically produce self-antigens or altered proteins in response to illness or damage, which activate dormant microglia. These latter go to areas of injury to capture dead cells and debris as the primary immunological effector [163]. More research is required to fully understand how microglia and astrocytes could cooperate with neurons in the AD environment.

## 5. Conclusions

Although our understanding of AD has significantly increased over the past few years, much more has to be discovered. The treatments for AD that are currently available only address symptoms rather than the underlying causes of the disease. Therefore, there is a critical need for therapeutic approaches that can engage the pathways underlying AD pathogenesis and impede the disease’s progression. One such downstream target for neurodegeneration that is a cause rather than an effect is neuroinflammation. Microglia are the primary regulators of neuroinflammation in the exceedingly complicated mechanisms associated with AD development. Additionally, microglia have a substantial role in causing synaptic dysfunction and loss, while the specific pathways are still unknown. Therefore, urgent progress in our understanding of the molecular and cellular mechanisms underlying the interaction between microglia and synapses is necessary for the creation of innovative anti-AD treatments. Additionally, more research is needed to determine whether or not preventing the loss of synapses by microglia reduces cognitive deficits and prevents neurodegeneration. New AD treatments and diagnostic options are now possible thanks to recent advances in our understanding of the main mechanisms underlying microglia failure in neurogenesis, plasticity modulation, and pruning. New therapy approaches for AD may result from focusing on these aberrant microglial mechanisms and reestablishing homeostasis. New biomarkers that indicate the function of various microglia are urgently required due to the complex and variety of the roles played by microglia in health and disease. Reactive microglia may play a crucial role in the early stages of AD progression and may lead to the discovery of early AD biomarkers because they have the ability to react and perceive their microenvironment. Microglia may be a possible pharmacological target to slow or stop the evolution of AD since they can interact with non-neuronal immune cells and change astrocyte activity. However, further research is needed to determine the precise functions of different reactive microglia subtypes in AD. Researchers are now able to investigate the functions of microglia in AD because of numerous technological advances. It is predicted that greater understanding of the roles of microglia in the onset and progression of AD would revive the interest of major pharmaceutical companies to reinvest in this research area and the creation of novel anti-AD therapeutic discoveries.

## Figures and Tables

**Figure 1 ijms-23-12990-f001:**
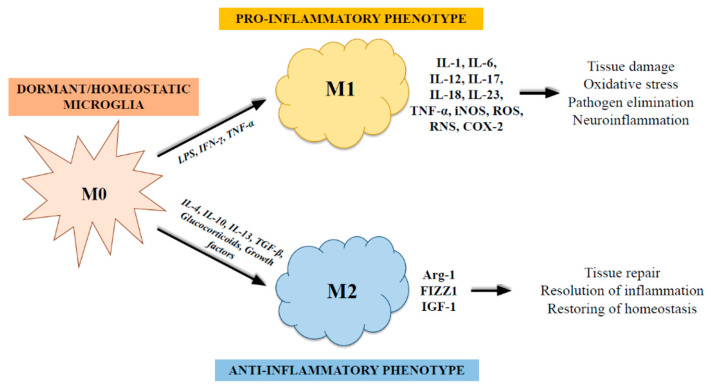
The two primary activation phenotypes of microglia. In physiological conditions, the microglia are dormant (M0). Injurious circumstances can cause the activation of resting microglia into distinct phenotypes, M1 and M2, having a proinflammatory phenotype (inducing neurological damage) and taking part in resolution of inflammation (with a neuroprotective role), respectively.

**Figure 2 ijms-23-12990-f002:**
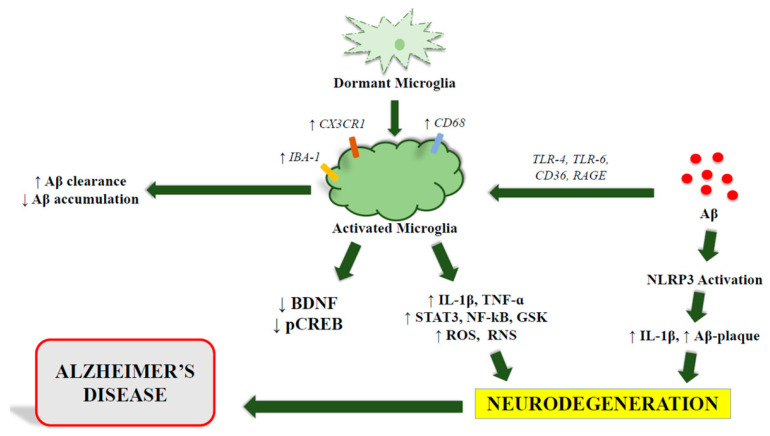
The role of microglia in AD. Microglia activation leads to the activation of various inflammatory signaling pathways. Its continuous stimulation can cause neurodegeneration.

## Data Availability

Not applicable.

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
