# Peer review of "Microglia and Alzheimer’s Disease"

_ijms, 2022, doi:10.3390/ijms232112990_

Round 1

Reviewer 1 Report

The manuscript by Merighi et al is a comprehensive and well-written review focusing on the role of microglia in Alzheimer's disease. I have only minor suggestion to improve the manuscript.

Authors review the litterature concerning the ability of microglia to adopt different phenotypes, protective or detrimental, in response to a variety of stimuli including signals not strictly correlated with the disease such as LPS and pesticides. It would be interesting to expand the section by adding information on the external and intrinsic signals causing a disfunctional reaction of microglia specifically during AD. Which are the proposed inciting molecules? where they come from? which is the role of astrocytes and adaptive immune cells?

Moreover, recent data obtained on the role played by TAM receptors in the detection and engulfment of amyloid beta plaques should be added and discussed. For example in the part of the manuscript where authors describe the protective role played by microglia in clearing the pathogenic amyloid beta.

Author Response

The manuscript by Merighi et al is a comprehensive and well-written review focusing on the role of microglia in Alzheimer's disease. I have only minor suggestion to improve the manuscript.

We appreciate the Reviewer's thoughtful criticism of our review and the chance to make it better. We have answered all the requests and new parts in the manuscript are written in red.

Authors review the litterature concerning the ability of microglia to adopt different phenotypes, protective or detrimental, in response to a variety of stimuli including signals not strictly correlated with the disease such as LPS and pesticides. It would be interesting to expand the section by adding information on the external and intrinsic signals causing a disfunctional reaction of microglia specifically during AD. Which are the proposed inciting molecules? where they come from? which is the role of astrocytes and adaptive immune cells?

We have included a paragraph under section 3 "Microglia in AD" that details how metals and oxidative stress both activate microglia specifically in AD. Additionally, we have added a section, 4. Microglia, astrocytes, and neurons crosstalk in AD, to emphasize and document this interaction in AD.

Moreover, recent data obtained on the role played by TAM receptors in the detection and engulfment of amyloid beta plaques should be added and discussed. For example in the part of the manuscript where authors describe the protective role played by microglia in clearing the pathogenic amyloid beta.

We thank the Reviewer for his/her indication. We have added and discussed the role played  by TAM receptors in the detection and engulfment of amyloid beta plaques in section 3 "Microglia in AD".

Reviewer 2 Report

In this manuscript, the author reviewed the important role of microglia in Alzheimer's disease, and discussed that microglia mediated inflammation may be an important target for treating Alzheimer's disease. There are several weaknesses in this study: 1) The interaction between microglia and other nerve cells is worth discussing; 2) The type of microglia activation in AD should be mentioned; 3) The pathways involved in neuroinflammation could be more comprehensive

Author Response

In this manuscript, the author reviewed the important role of microglia in Alzheimer's disease, and discussed that microglia mediated inflammation may be an important target for treating Alzheimer's disease. 

We appreciate the Reviewer's thoughtful criticism of our review and the chance to make it better. 

We have answered all the requests and new parts in the manuscript are written in red.

There are several weaknesses in this study: 

1) The interaction between microglia and other nerve cells is worth discussing; 

We thank the Reviewer for his/her indication. Therefore, we have added a section, 4. Microglia, astrocytes, and neurons crosstalk in AD, to emphasize and document interaction between microglia and other nerve cells in AD.

2) The type of microglia activation in AD should be mentioned; 

According to the Reviewer’s indication, we have mentioned and specified the type of microglia activation in AD, in section 3. Microglia in AD.

3) The pathways involved in neuroinflammation could be more comprehensive

We thank the Reviewer for his/her useful suggestion. We have added a new part in section 3.Microglia in AD concerning the role of NLRP3 and TAM in microglia and